

# Gaussian-state Ansatz for the non-equilibrium dynamics of quantum spin lattices

**Raphaël Menu[1⋆] and Tommaso Roscilde[2]**

**1** Theoretische Physik, Universität des Saarlandes, D-66123 Saarbrücken, Germany
**2** Univ Lyon, ENS de Lyon, CNRS, Laboratoire de Physique, F-69342 Lyon, France

⋆ raphael.menu@physik.uni-saarland.de

## Abstract

The study of non-equilibrium dynamics is one of the most important challenges of modern quantum many-body physics, in relationship with fundamental questions in quantum statistical mechanics, as well as with the fields of quantum simulation and computing. In this work we propose a Gaussian Ansatz for the study of the nonequilibrium dynamics of quantum spin systems. Within our approach, the quantum spins are mapped onto Holstein-Primakoff bosons, such that a coherent spin state – chosen as the initial state of the dynamics – represents the bosonic vacuum. The state of the system is then postulated to remain a bosonic Gaussian state at all times, an assumption which is exact when the bosonic Hamiltonian is quadratic; and which is justified in the case of a nonlinear Hamiltonian if the boson density remains moderate. We test the accuracy of such an Ansatz in the paradigmatic case of the $S = 1/2$ transverse-field Ising model, in one and two dimensions, initialized in a state aligned with the applied field. We show that the Gaussian Ansatz, when applied to the bosonic Hamiltonian with nonlinearities truncated to quartic order, is able to reproduce faithfully the evolution of the state, including its relaxation to the equilibrium regime, for fields larger than the critical field for the paramagnetic-ferromagnetic transition in the ground state. In particular the spatio-temporal pattern of correlations reconstructed via the Gaussian Ansatz reveals the dispersion relation of quasiparticle excitations, exhibiting the softening of the excitation gap upon approaching the critical field. Our results suggest that the Gaussian Ansatz correctly captures the essential effects of nonlinearities in quantum spin dynamics; and that it can be applied to the study of fundamental phenomena such as quantum thermalization and its breakdown.

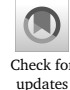

# 1   Introduction

The non-equilibrium unitary dynamics of quantum many-body systems represents a central topic of modern quantum physics: it lies at the core of the coherent manipulation of complex quantum states with quantum devices [1–4]; and it represents the mechanism by which equilibration and the emergence of statistical ensembles occurs [5–10]. In the case of systems with a time-independent Hamiltonian (which will be the focus of this work), central questions concern the propagation of correlations and entanglement, and the scrambling of quantum information [4, 11]; how such phenomena manifest the nature of elementary excitations [12–16]; and how they lead to the onset of equilibration [7, 8, 10]. Answering to the above questions quantitatively for systems of increasing size is a significant challenge, due to the exponential increase of the Hilbert-space dimensions with system size, making an exact numerical treatment impractical for systems going beyond *e.g.* a few tens of qubits. The development of efficient numerical schemes which *approximate* the quantum many-body evolution is therefore the only realistic strategy to approach the problem theoretically – the only alternative solution comes from the direct experimental implementation of the dynamics of interest via an analog or digital quantum simulator [2]. By "efficient scheme" we mean here a numerical approach whose computational cost scales polynomially with system size; and which ideally can be improved systematically, while maintaining a polynomial cost.

A general framework for the development of efficient numerical schemes is the one of wavefunction Ansätze, positing that the state vector has an explicit functional form depending on a number of variational parameters which is much smaller than the Hilbert-space dimen-

sions. Relevant examples of variational Ansätze for the unitary evolution are tensor network states [17, 18], neural-network quantum states [19], Jastrow-like wavefunctions [20–22], etc. Among these families of states, the tensor-network states can be systematically improved by increasing the bond dimension, whose logarithm regulates the maximum amount of subsystem entanglement entropy that the state can accommodate. Nonetheless unitary evolutions lead systematically to extensive entanglement entropies, and to the so-called entanglement barrier [23]; namely the exponential increase of the bond dimension required to reproduce the state faithfully. Other families of states allow for systematical improvements without facing an entanglement barrier nonetheless, all these approaches are in general very demanding from a computational point of view, and their computational cost further scales significantly with the dimensions of the local Hilbert space describing individual degrees of freedom. An alternative to wavefunction Ansätze, valid for spin or bosonic systems, is offered by the discrete truncated Wigner approximation (DTWA). This approach postulates an evolved state of the system expressed in terms of the Wigner-function representation of the initial state, and of phase-point operators dependent on parameters evolved according to classical equations of motion [24]. The DTWA approach is very powerful in representing evolutions arbitrarily far from equilibrium [25], as it is based on a Monte Carlo sampling of classical trajectories initialized via the Wigner function of the initial state, without making any assumption on the statistics of quantum fluctuations. Yet it represents instantaneous expectation values as incoherent Monte Carlo averages, and therefore it misses many-body interference effects which are at the basis of fundamental phenomena (such as *e.g.* many-body localization [26]).

In this work we propose and validate an alternative approximation schemes for quantum evolutions of generic quantum systems – bosonic or fermionic alike – based on Gaussian states. Gaussian many-body states have the property that the reduced density matrix of each subsystem has a Gaussian form, such that all observables obey Wick's theorem – namely they can be reconstructed from the knowledge of the average fields and the covariance matrix, whose elements are given by the regular and anomalous two-point Green's function. Hence reconstructing the evolution of the state amounts to tracking the dynamics of the average fields and covariance matrix, obeying coupled non-linear equations of motion. This scheme is well known for fermionic (bosonic) systems as the time-dependent Hartree-Fock (Hartree-Fock-Bogolyubov) approach; yet in this work we propose its application to quantum spin systems, specifically based on the Holstein-Primakoff (HP) spin-boson mapping. A Gaussian Ansatz for the equilibrium state of the non-linear bosonic Hamiltonian resulting from the HP mapping is at the core of the so-called modified spin-wave theory [27]. Our approach can therefore be viewed as a time-dependent version of modified spin-wave theory. On the other hand, conventional time-dependent spin-wave theory amounts to discarding all non-linear terms in the bosonic Hamiltonian. The latter approach has been successfully applied to a large variety of non-equilibrium problems [14, 28–32, 46], but it has also significant limitations, as we shall argue in this work.

We apply the Gaussian Ansatz approach to the non-equilibrium evolution of a paradigmatic model for quantum simulation, namely the $S = 1/2$ transverse-field Ising model (TFIM), initialized in a state aligned with the applied field. This dynamics corresponds to a quantum quench starting from the ground state at infinite field, and triggered by changing the field abruptly to a finite value. We first benchmark our approach in the case of the one-dimensional TFIM, which can be exactly solved by mapping it onto free fermions. Our results show that the dynamics of the free-fermion system can be mimicked quantitatively by that of a system of strongly interacting Holstein-Primakoff bosons, for quenches to fields as low as the critical one for the ground-state phase transition from paramagnet to ferromagnet. Even more convincing results are obtained for the case of the two-dimensional TFIM, which cannot be solved exactly; but whose dynamics can be quantitatively reconstructed via time-dependent

variational calculations [33]. In both cases (1d and 2d) we show that the Gaussian Ansatz is able to capture fundamental features of the spatio-temporal structure of the quantum correlations developing after the quench, as observed in their Fourier transform (via a so-called quench spectroscopy scheme [14–16]), which allows one to reconstruct the dispersion relation of elementary excitations. The dispersion relation reconstructed via quench spectroscopy is in very good agreement with the best available benchmark results down to the critical field, and it captures in particular the softening of the excitation gap upon approaching the critical field, suggesting that the non-equilibrium dynamics is sensitive to ground-state quantum criticality.

Our paper is structured as follows: Sec. 2 introduces the spin-boson mapping and the Gaussian-Ansatz approach to the resulting nonlinear bosonic Hamiltoinian; Sec. 3 discusses the results for the 1d TFIM, while Sec. 3.2.2 presents the results for the 2d TFIM. Conclusions and pespectives are offered in Sec. 5.

## 2 Gaussian-state Ansatz for quantum spin systems

In this section we illustrate the spin-boson mapping and the Gaussian-Ansatz (GA) approach which allows us to cast the evolution of the system in terms of the evolution of the average fields and covariance matrix for the bosonic operators.

### 2.1 Spin-to-boson mapping

We shall consider a general linear/bilinear spin Hamiltonian, with the form

$$\hat{\mathcal{H}} = \frac{1}{2} \sum_{\mu,\nu=x,y,z} \sum_{i,j} J_{ij}^{\mu\nu} \hat{S}_i^{\mu} \hat{S}_j^{\nu} - \sum_i \mathbf{H}_i \cdot \hat{\mathbf{S}}_i \,, \tag{1}$$

where $\hat{S}_i^{\mu}$ ($\mu = x, y, z$) are quantum spin operators of length $S$ ($|\hat{\mathbf{S}}_i|^2 = S(S+1)$) attached to the $i$-th site of an arbitrary lattice; $J_{ij}^{\mu\nu}$ is the coupling matrix; and $\mathbf{H}_i$ is a local magnetic field.

The choice of the quantization axis (hereafter denoted as $z$) on every lattice site is dictated by the initial state of the evolution, which is assumed here to be a *coherent spin state* aligned with the quantization axis

$$|\psi(0)\rangle = \otimes_{i=1}^N |m = S\rangle_i \,, \tag{2}$$

where $\hat{S}_i^z |m\rangle_i = m |m\rangle_i$.

The above choice is by no means restrictive in terms of the orientations of the single-site states, as any initial state composed of initially polarized spins (albeit along arbitrary local directions) can be mapped via local rotations to the state of Eq. (2). If the local rotations are then applied to transform a linear/bilinear spin Hamiltonian, its form will be generically that of Eq. (1).

The choice of the quantization axis based on the local orientations of the spins in the initial state is crucial to define the Holstein-Primakoff (HP) mapping of spins onto bosons:

$$\hat{S}_i^z = S - \hat{n}_i \,, \tag{3a}$$

$$\hat{S}_i^+ = \sqrt{2S - \hat{n}_i} \, \hat{b}_i \,, \tag{3b}$$

$$\hat{S}_i^- = \hat{b}_i^\dagger \sqrt{2S - \hat{n}_i} \,, \tag{3c}$$

where $\hat{b}_i, \hat{b}_i^\dagger$ are bosonic operators, satisfying the constraint $0 \le \hat{n}_i = \hat{b}_i^\dagger \hat{b}_i \le 2S$ to preserve the dimensions of the Hilbert space for quantum spins. In the following this constraint will only be retained on average, namely we shall only consider evolutions which respect the inequality $\langle \hat{b}_i^\dagger \hat{b}_i \rangle \le 2S$ at all times. By construction, the initial state Eq. (2) is the bosonic *vacuum* $|\psi(0)\rangle = \otimes_{i=1}^N |\emptyset\rangle_i$, which is a Gaussian state.

Under the HP mapping, and upon expanding $\sqrt{2S - \hat{n}_i} = 2S(1 - \hat{n}_i/(4S) + ...)$, the spin Hamiltonian takes the generic nonlinear form

$$\hat{\mathcal{H}} = \hat{\mathcal{H}}_0 + \hat{\mathcal{H}}_1 + \hat{\mathcal{H}}_2 + \hat{\mathcal{H}}_3 + \hat{\mathcal{H}}_4 + ..., \tag{4}$$

where $\hat{\mathcal{H}}_n$ is a bosonic Hamiltonian containing only products of $n$ bosonic operators. The zero-th order term is a constant, representing the mean-field energy if the initial state is the ground state of the system within the mean-field approximation (as it will be the case in the examples offered in this work). The linear term

$$\hat{\mathcal{H}}_1 = \sum_i (\gamma_i \hat{b}_i + \gamma_i^* \hat{b}_i^\dagger) \tag{5}$$

displaces the vacuum, and it can be eliminated via a shift of the bosonic operators; while the quadratic term

$$\hat{\mathcal{H}}_2 = \sum_{ij} \left[ \mathcal{A}_{ij} \hat{b}_i^\dagger \hat{b}_j + (\mathcal{B}_{ij} \hat{b}_i \hat{b}_j + \text{h.c.}) \right] \tag{6}$$

is the basis of linear spin-wave (LSW) theory. When truncating the Hamiltonian to quadratic order, the initial bosonic vacuum transforms into a displaced and squeezed vacuum, namely it remains a Gaussian state.

## 2.2 Gaussian-state Ansatz

The central assumption of the GA approach is that, even when retaining nonlinear terms beyond quadratic, the bosonic state remains Gaussian. For practical purposes we will hereafter restrict our attention to quartic Hamiltonians, namely we shall discard all terms $\hat{\mathcal{H}}_n$ with $n \geq 5$. This approximation is valid if $\langle n_i \rangle/(2S) \ll 1$; and the same assumption underpins the validity of the Gaussian Ansatz, which is explicitly justified when the nonlinear effects beyond LSW theory are moderate.

The evolved state, assumed to be Gaussian, has the property of satisfying Wick's theorem at all times. This implies that, considering the shifted operators $a_i = b_i - \langle b_i \rangle$, the $2n$-point correlation function breaks down to the sum of products of two-point correlation functions:

$$\langle \hat{a}_{i_1}^{d_1} \hat{a}_{i_2}^{d_2} ... \hat{a}_{i_{2n-1}}^{d_{2n-1}} \hat{a}_{i_{2n}}^{d_{2n}} \rangle =: \sum_p \left\langle \hat{a}_{p(i_1)}^{d_{p(1)}} \hat{a}_{p(i_2)}^{d_{p_2}} \right\rangle ... \left\langle \hat{a}_{p(i_{2n-1})}^{d_{p(2n-1)}} \hat{a}_{p(i_{2n})}^{d_{p(2n)}} \right\rangle, \tag{7}$$

where the symbols $d$ can be 1 or $\dagger$, and the sum runs on $(2n-1)!!$ ordered permutations, in which each bosonic operator is paired with another operator to its right.

Positing the validity of the above Wick's decomposition of $2n$-point correlation functions implies that all the observables of interest can be reconstructed from the knowledge of the regular and anomalous Green's functions, $G_{ij} = \langle \hat{a}_i^\dagger \hat{a}_j \rangle$ and $F_{ij} = \langle \hat{a}_i \hat{a}_j \rangle$, respectively. Hence studying the evolution of a Gaussian state amounts to monitoring the evolution of the average field values $\beta_i = \langle \hat{b}_i \rangle$ and of the elements of the so-called covariance matrix $\langle \hat{a}_i^{d_i} \hat{a}_j^{d_j} \rangle$, namely solving $\mathcal{O}(N^2)$ coupled non-linear differential equations

$$\frac{\mathrm{d}}{\mathrm{d}t} \beta_i = i \langle [\hat{\mathcal{H}}, \hat{b}_i] \rangle = \mathcal{E}_{\beta_i}(\{\beta_l, G_{lm}, F_{lm}\}), \tag{8a}$$

$$\frac{\mathrm{d}}{\mathrm{d}t} G_{ij} = i \langle [\hat{\mathcal{H}}, \hat{a}_i^\dagger \hat{a}_j] \rangle = \mathcal{E}_{G_{ij}}(\{\beta_l, G_{lm}, F_{lm}\}), \tag{8b}$$

$$\frac{\mathrm{d}}{\mathrm{d}t} F_{ij} = i \langle [\hat{\mathcal{H}}, \hat{a}_i \hat{a}_j] \rangle = \mathcal{E}_{F_{ij}}(\{\beta_l, G_{lm}, F_{lm}\}), \tag{8c}$$

where $\mathcal{E}$'s are non-linear functions to be determined. Here and in the following we set $\hbar = 1$.

### 2.3 Application to the transverse-field Ising model

For the rest of this paper we shall specialize our attention to the paradigmatic case of transverse-field Ising models, whose Hamiltonian reads:

$$\hat{\mathcal{H}} = -\frac{1}{2}\sum_{i,j} J_{ij}\hat{S}_i^x\hat{S}_j^x - \Omega\sum_i \hat{S}_i^z, \tag{9}$$

where $J_{ij}$ is the coupling and $\Omega$ is the transverse field. This model describes the magnetic behavior of magnetic insulators in an external field [34,35], as well as the many-body physics of quantum simulators based on trapped ions [36], Rydberg atoms [37–39], or superconducting circuits [40] to cite a few relevant examples. In the case of ferromagnetic interactions $J_{ij} \geq 0$, the equilibrium phase diagram of this model exhibits generically a quantum phase transition at a critical field value $\Omega_c$ dividing a large-field paramagnetic phase (not breaking any symmetry) from a small-field ferromagnetic phase (breaking the $\mathbb{Z}_2$ symmetry of the spin Hamiltonian in the thermodynamics limit).

The choice of the quantization axis along the field immediately suggests that we shall specialize our attention to evolutions initialized in the state aligned with the field itself. Performing the HP transformation and retaining bosonic non-linearities up to quartic terms only, one obtains the following bosonic Hamiltonian

$$\hat{\mathcal{H}} \simeq -N\Omega S - \frac{S}{4}\sum_{i,j} J_{ij}(\hat{b}_i^\dagger\hat{b}_j + \hat{b}_i^\dagger\hat{b}_j^\dagger + \text{h.c.}) + \Omega\sum_i \hat{b}_i^\dagger\hat{b}_i$$
$$+ \frac{1}{8}\sum_{i<j} J_{ij}\left[(\hat{b}_i^\dagger + \hat{b}_i)(\hat{b}_j^\dagger\hat{b}_j^\dagger\hat{b}_j + \hat{b}_j^\dagger\hat{b}_j\hat{b}_j) + \text{h.c.}\right], \tag{10}$$

which includes terms that describe the pair creation, destruction or hopping of bosons, and terms describing hopping conditioned on the local density of particles or on pair creation/destruction. The Hamiltonian contains only even terms in the bosons, such that the initial bosonic vacuum is not displaced by the evolution, namely $\beta_i = \langle\hat{b}_i\rangle = 0$ and $\hat{a}_i = \hat{b}_i$. This means that at the mean-field level there is absolutely no dynamics; and that the quantum dynamics comes exclusively from the establishment of entanglement among the spins, which is captured at the level of the GA via the evolution of the covariance matrix. The equations of motion for the covariance matrix are reported in App. A.

### 2.4 Linear spin-wave theory and dynamical instability

It is very instructive to explicitly examine the quantum dynamics predicted by linear spin-wave (LSW) theory, which amounts to retaining only the quadratic terms in the bosonic Hamiltonian of Eq. (10). For periodic lattices, the latter Hamiltonian can be Bogolyubov-diagonalized in momentum space. Introducing the Fourier-transformed bosonic operators $\hat{b}_\mathbf{k} = N^{-1/2}\sum_i e^{-i\mathbf{k}\cdot\mathbf{r}_i}\hat{b}_i$, one obtains

$$\hat{\mathcal{H}}_0 + \hat{\mathcal{H}}_2 = E_{\text{MF}} + \sum_\mathbf{k}\left[A_\mathbf{k}\hat{b}_\mathbf{k}^\dagger\hat{b}_\mathbf{k} + \frac{1}{2}\left(B_\mathbf{k}\hat{b}_\mathbf{k}\hat{b}_{-\mathbf{k}} + \text{h.c.}\right)\right] = E_0 + \sum_\mathbf{k}\omega_\mathbf{k}\hat{\alpha}_\mathbf{k}^\dagger\hat{\alpha}_\mathbf{k}, \tag{11}$$

where $E_{\text{MF}} = -N\Omega S$ is the mean-field energy; $E_0 = E_{\text{MF}} - 1/2\sum_\mathbf{k}(A_\mathbf{k} - \omega_\mathbf{k})$ is the spin-wave energy; the $A$ and $B$ coefficients read

$$A_\mathbf{k} = -\frac{J_\mathbf{k}S}{2} + \Omega, \qquad B_\mathbf{k} = -\frac{J_\mathbf{k}S}{2}, \tag{12}$$

with $J_\mathbf{k} = \sum_\mathbf{r} e^{-i\mathbf{k}\cdot\mathbf{r}}J_{i,i+\mathbf{r}}$ the Fourier transform of the Ising couplings;

$$\omega_\mathbf{k} = \sqrt{A_\mathbf{k}^2 - B_\mathbf{k}^2} = \sqrt{\Omega(\Omega - J_\mathbf{k}S)} \tag{13}$$

is the dispersion relation of the Bogolyubov quasi-particles, described by the operators $\hat{\alpha}_{\mathbf{k}} = u_{\mathbf{k}}\hat{b}_{\mathbf{k}} + v_{\mathbf{k}}\hat{b}_{-\mathbf{k}}^{\dagger}$ with $u_{\mathbf{k}} = \sqrt{1/2(A_{\mathbf{k}}/\omega_{\mathbf{k}}+1)}$ and $v_{\mathbf{k}} = \mathrm{sign}(B_{\mathbf{k}})\sqrt{1/2(A_{\mathbf{k}}/\omega_{\mathbf{k}}-1)}$.

Clearly LSW theory exhibits an instability when $\Omega < \max_{\mathbf{k}} J_{\mathbf{k}}S$, which implies that one of the spin-wave eigenfrequencies of Eq. (13) becomes purely imaginary. In the case of nearest-neighbor ferromagnetic couplings (with strength $J$) on a hyper-cubic lattice in $d$ spatial dimensions, one has $J_{\mathbf{k}} = 2J\sum_{i=1}^{d}\cos(k_i)$, so that the instability condition is $\Omega < \Omega^* = 2SdJ$ at which the eigenmode at $\mathbf{k} = 0$ becomes unstable. This implies that the linearized dynamics is intrinsically limited to large fields. As we will see in the next section, this fundamental limitation is removed when including the first nonlinear term in the Hamiltonian within the GA approach.

### 2.5 *Intermezzo*: Modified spin-wave theory vs. dynamical Gaussian Ansatz

Before concluding this section, we briefly review Takahashi's modified spin-wave (MSW) theory [27], which applies the notion of Gaussian Ansatz to the study of the equilibrium properties of the system. In its ground-state formulation for the transverse-field Ising model, MSW theory amounts to finding the Gaussian state which minimizes the expectation value of the non-linear Hamiltonian $\langle\hat{\mathcal{H}}\rangle$. To perform the minimization, the Gaussian state must be represented in terms of independent parameters: in the case of translationally invariant systems, a convenient parametrization is obtained by representing the state via its density matrix as $\rho = \exp(-\sum_{\mathbf{k}}\tilde{\omega}_{\mathbf{k}}\hat{\alpha}_{\mathbf{k}}^{\dagger}\hat{\alpha}_{\mathbf{k}}/T)/\mathcal{Z}$; $T$ is the temperature (to be set to zero when looking at the ground state physics); $\hat{\alpha}_{\mathbf{k}} = \cosh\theta_{\mathbf{k}}\hat{b}_{\mathbf{k}} + \sinh\theta_{\mathbf{k}}\hat{b}_{-\mathbf{k}}^{\dagger}$ are Bogolyubov-transformed operators; and $\mathcal{Z}$ is the partition function. The above form of the density matrix embodies explicitly the Gaussian Ansatz, namely the state is the exponential of a generic quadratic form in the boson operators, here represented already in its Bogolyubov-diagonalized form delivering the eigenfrequencies $\tilde{\omega}_{\mathbf{k}}$. Please notice that the coefficients of the Bogolyubov transformation and the eigenfrequencies $\tilde{\omega}_{\mathbf{k}}$ differ from those introduced in the previous section (Sec. 2.4) within LSW theory, due to the inclusion of the nonlinearities.

The ground state energy (or more generally the finite-temperature free energy) is then variationally minimized with respect to the $\theta_{\mathbf{k}}$ angles. This leads to a set of coupled non-linear equations whose solution provides the best self-consistent Gaussian approximation to the ground state. It often happens that the coupled non-linear equations do not admit a solution [41]: this is for instance exhibited by the transverse-field Ising model in the vicinity of the critical field $\Omega_c$. The loss of a solution for MSW theory suggests that quantum fluctuations become so strong that their self-consistent harmonic treatment – underlying the image of the Gaussian Ansatz – is no longer applicable. Nonetheless the non-equilibrium evolution of the Gaussian-state Ansatz does *not* suffer from this problem, as its solution is mathematically always well defined for any value of the Hamiltonian parameters. The main pathology that the GA approach can suffer from, is the uncontrolled proliferation of bosons beyond the maximum allowed value; namely the fact that, for times later than a given time $t^*$, $\langle\hat{n}_i\rangle(t) > 2S$. After the time $t^*$ the physics of the bosonic system would depart from that of the spin system, and the approach is no longer predictive. This pathology is clearly encountered by LSW theory for $\Omega < \Omega^*$ (see Sec. 2.4); yet, as already mentioned above, the inclusion of non-linearities removes this problem within the GA approach for all the evolutions studied in this work.

## 3 A solvable case: The transverse-field Ising chain

We shall first benchmark the GA approach for the exactly solvable case of the $S = 1/2$ one-dimensional TFIM with nearest-neighbor interactions, namely Eq. (9) with $J_{ij} = J\delta_{j,i+1}$ de-

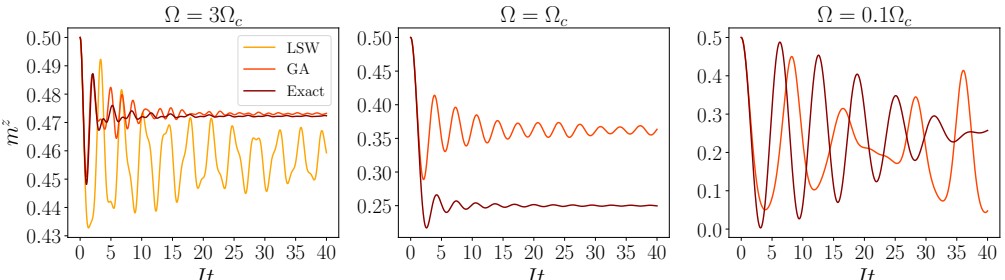

Figure 1: Evolution of the transverse magnetization $m^z$ of the 1d TFIM after a quantum quench to three field values $\Omega = 3\Omega_c$, $\Omega_c$ and $0.1\Omega_c$. All the data refer to a chain of $N = 50$ spins with periodic boundary conditions. The leftmost panel compares the GA results with the LSW and the exact ones; the other panels only report the GA and the exact results.

fined on a periodic chain. The one-dimensional TFIM can be exactly solved by mapping spins onto spinless fermions via the non-local Jordan-Wigner transformation [42, 43]. The density of Jordan-Wigner fermions corresponds to the deviation of the spins from the configuration fully polarized with the field, $\hat{S}^z = 1/2 - \hat{f}_i^\dagger \hat{f}_i$ – where $\hat{f}_i, \hat{f}_i^\dagger$ are fermionic operators. The transverse spin components are instead related in a non-local way to the fermionic operators ($\hat{S}_i^+ = \hat{f}_i \exp(i\pi \sum_{j<i} \hat{f}_i^\dagger \hat{f}_i)$) in order for the fermionic anticommutation relations to hold. Under the Jordan-Wigner transformation, the spin Hamiltonian becomes a quadratic fermionic one

$$\mathcal{H} = -\frac{J}{4}\sum_i \left(\hat{f}_i^\dagger \hat{f}_{i+1}^\dagger + \hat{f}_i^\dagger \hat{f}_{i+1} + \text{h.c.}\right) - \Omega \sum_i (1/2 - \hat{f}_i^\dagger \hat{f}_i) \,. \tag{14}$$

Hence, within the fermionic picture, the dynamics initialized in the state aligned with the field corresponds to the evolution of a gas of spinless fermions initialized in their vacuum state, and evolving under a dynamics of pair creation/annihilation on neighboring sites, as well as hopping, in a chemical potential dictated by $-\Omega$. A Bogolyubov diagonalization of the fermionic operators leads to the prediction of a ground-state phase transition occurring for a field $\Omega_c = J/2$ between a paramagnetic phase for $\Omega > \Omega_c$ and a ferromagnetic phase for $\Omega < \Omega_c$. The spinless fermions onto which the spins are mapped can be Bogolyubov diagonalized to give the dispersion relation [43]

$$\epsilon_k = \sqrt{\Omega(\Omega - J\cos k) + (J/2)^2}\,, \tag{15}$$

which becomes gapless for $k = 0$ at $\Omega = \Omega_c$. This dispersion relation clearly differs from that of the linearized bosons (Eq. (13)). Nonetheless the bosonic and fermionic populations should be identical (when solving the bosonic problem exactly, beyond LSW theory). Therefore the bosonic approach to the 1d TFIM amounts to reproducing the physics of non-interacting spinless fermions in terms of the dynamics of strongly interacting bosons. It is rather obvious that accounting for the non-linearities in the bosonic problem – attempted in this work within the GA approach – is an essential ingredient for this endeavor to be successful.

The solvable 1d TFIM clearly offers a very valuable reference for approximate methods such as the GA approach. At the same time, the dynamics of an integrable system such as the 1d TFIM is highly non-generic, and it poses a significant challenge to any approximation scheme. Indeed the dynamics is strongly dominated by a feature – the presence of an extensive number of conserved quantities – which can only be captured by the exact treatment of the problem.[1] In

---

[1]LSW theory possesses an extensive number of conserved quantities (the populations $\hat{\alpha}_\mathbf{k}^\dagger \hat{\alpha}_\mathbf{k}$) which nonetheless do not correspond to the correct conserved quantities of the fermionic theory. Given its non-linear nature, it is unclear whether the GA approach possesses many conserved quantities beyond the energy.

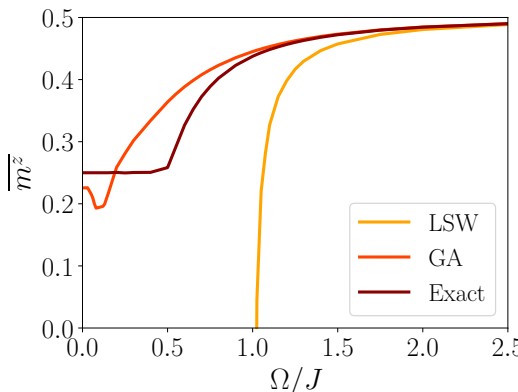

Figure 2: Time-averaged transverse magnetization along the quench dynamics of the 1d TFIM, comparing the GA, LSW and exact values. The time window for the average is $\tau J = 40$. All the other parameters as in Fig. 1.

the following we will focus on a few selected aspects of the dynamics, revealing the emergence of a generalized Gibbs ensemble [9] in the long-time dynamics of the transverse magnetization; and fundamental spectral features from the Fourier analysis of the spatio-temporal correlation pattern.

## 3.1 Transverse magnetization

The transverse magnetization $m^z = \frac{1}{N}\sum_i \langle S_i^z \rangle = S - \frac{1}{N}\sum_i G_{ii}$ is a measure of the density of the interacting gas of HP bosons (see Eq. (3a)). Monitoring its dynamics offers a fundamental test of the hypothesis of diluteness of HP bosons, which is at the core of the Taylor expansion of the square roots in the HP transformation; and which offers the best conditions for the GA approach to be successful. Here we shall probe to what extent the GA can cope with the proliferation of bosons and the corresponding increase of non-linear effects.

Starting from the bosonic vacuum – namely the ground state at $\Omega = \infty$ – as initial state, the diluteness condition should be met at best when quenching the field $\Omega$ to large values, $\Omega \gg J$. The dynamics of the transverse magnetization is displayed on Fig 1 for three field values ($\Omega = 3\Omega_c, \Omega_c, 0.1\Omega_c$). In the case of a large field $\Omega = 3\Omega_c$ we observe that the agreement of the GA with the exact solution is rather remarkable. For that field value we can also perform a LSW calculation, since LSW theory becomes unstable only for $\Omega < \Omega^* = J = 2\Omega_c$. The comparison with the GA result and the exact one shows that nonlinear effects beyond LSW theory are already sizable, in spite of the fact that the boson density is only $\langle n \rangle \sim 0.03$: LSW theory predicts a boson population which is significantly larger, with much larger fluctuations and with a dominant frequency which is about half of that exhibited by the exact solution. When quenching to a smaller field $\Omega = \Omega_c$, we lose the LSW solution ($\Omega < \Omega^*$) because of the uncontrolled proliferation of bosons. On the other hand the GA approach still gives a stable solution, which underestimates the boson population generated by the quench, while still capturing the right dominant frequency of oscillations and the presence of damping in the oscillations. For an even larger quench to $\Omega = 0.1\Omega_c$, the GA approach agrees with the exact solution only at short times, while it clearly deviates significantly from the exact results at longer times, albeit exhibiting large oscillations similar to those of the exact solution. It is very important to remark here that the boson population is found to rise to densities $\langle n \rangle \approx 0.45$, which can no longer be considered small with respect to their maximum value $n_{max} = 1$ allowed by the spin-to-boson mapping. This means that not only the assumption of a Gaussian state is to be

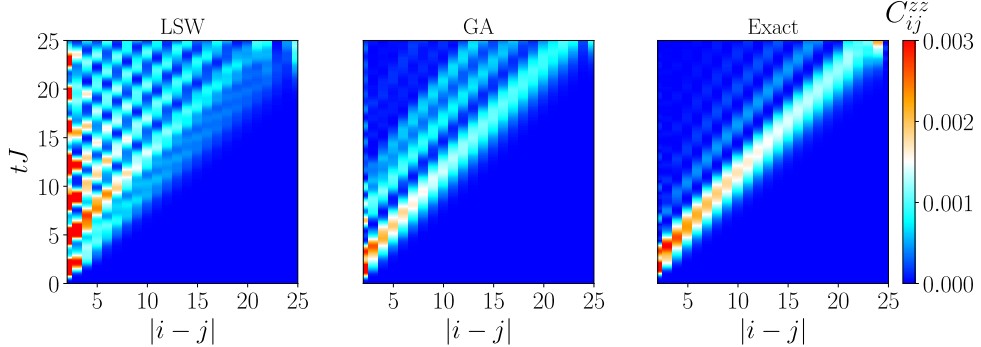

Figure 3: Evolution of the spin-spin correlation function for the $z$ component of spins in the 1d TFIM: (a-c) false color plot for the dynamics of $C^{zz}(i-j;t)$ as predicted by LSW theory, GA approach, and the exact solution. All the data refer to a quench to the field value $\Omega = 3\Omega_c$ for a chain of length $N = 50$.

called into question, but also the truncation of the bosonic Hamiltonian to the lowest nonlinear order is no longer justified. Hence it is imaginable that pushing the bosonic Hamiltonian to higher orders of expansion would lead to a significantly better agreement – although the calculations would become rather cumbersome.

A summarizing picture of the dynamics of the magnetization is offered in Fig. 2, showing the time-averaged magnetization $\overline{m^z} = \frac{1}{\tau}\int_0^\tau dt\; m^z(t)$ (with $\tau J = 40$) as a function of the quench field $\Omega$. We first analyze the exact result, which clearly exhibits a behavior incompatible with standard thermalization. Indeed, if the eigenstate thermalization hypothesis applied [8], the time-averaged magnetization would be expected to converge to the thermal average value within the Gibbs ensemble, at a temperature $T$ such that $\langle \hat{\mathcal{H}} \rangle_T = \langle \psi(0)|\hat{\mathcal{H}}|\psi(0)\rangle$. This is clearly not the case for $\Omega < \Omega_c$: even though the applied field decreases below $\Omega_c$, the time-averaged magnetization remains locked to the value $\approx 0.25$. This non-thermal behavior is clearly due to the non-integrable nature of the TFIM at finite field; and also to the fact that the $\Omega = 0$ limit is equally special, given that it corresponds to another integrable limit in which all individual spin operators $\hat{S}_i^x$ commute with the Hamiltonian.

Fig. 2 compares the exact result with the prediction of the GA approach as well as with that of LSW theory. The two approaches match with the exact result for high fields $\Omega \gg \Omega_c$; yet the LSW results start deviating from the exact ones when $\Omega$ approaches $\Omega^* = 2\Omega_c$ from above, at which LSW theory develops its instability and stops being predictive. On the other hand the GA approach remains quantitative for all values of $\Omega$, although the agreement with the exact result deteriorates upon approaching $\Omega_c$. We reiterate the fact that this deviation may be due to the failure of the Gaussian approximation as well as to the truncation of the non-linearities in the bosonic Hamiltonian. In particular the GA approach appears to capture the salient feature of the non-thermal behavior of the system, namely the fact that $\overline{m^z}$ does not vanish even when $\Omega \to 0$.

So far we have dealt with a single-spin property (the average spin polarization); in the next section we shall explore instead the dynamics of correlations, and how this dynamics reveals fundamental features of the excitation spectrum.

## 3.2 Dynamics of correlations and quench spectroscopy

The dynamics of two-point correlations is one of the most insightful aspects of the non-equilibrium evolution of quantum many-body systems. Indeed, in systems with elementary excitations having the nature of well-defined quasi-particles, the development of correlations starting *e.g.* from an uncorrelated state proceeds via the propagation of such quasi-particles;

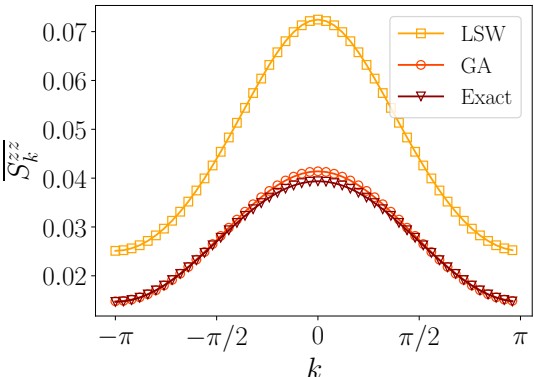

Figure 4: Time-averaged instantaneous structure factor $\overline{S_k^{zz}}$. All the data refer to a quench to the field value $\Omega = 3\Omega_c$ for a chain of length $N = 50$.

and it possesses a causal light-cone structure [44, 45] revealing the existence of a maximum speed of propagation for the excitations. A more detailed analysis of the structure of correlations within the light-cone allows one to inspect the excitation spectrum of the system via the so-called quench spectroscopy scheme [14–16]. In the following we shall consider the time evolution of the spin-spin correlation function

$$C^{\mu\mu}(i,j;t) = \langle \hat{S}_i^\mu \hat{S}_j^\mu \rangle - \langle \hat{S}_i^\mu \rangle \langle \hat{S}_j^\mu \rangle \,, \tag{16}$$

focusing on the case $\mu = z$ and $\mu = x$.

### 3.2.1 $\langle S^z S^z \rangle$ correlations and structure factor

We first focus on the spin-spin correlation function for the $z$ spin components. This quantity is easily accessible via the exact solution of the problem using fermionization, as it corresponds to the density-density correlation function for the fermions, $C^{zz}(i,j;t) = \langle (1/2 - \hat{f}_i^\dagger \hat{f}_i)(1/2 - \hat{f}_j^\dagger \hat{f}_j) \rangle$. The GA expression for this quantity is instead $C^{zz}(i,j;t) = G_{ij}(\delta_{ij} + G_{ij}^*) + |F_{ij}|^2$; the same expression can be used as well within LSW theory, but evolving the $G$ and $F$ functions by using linearized equations of motion. Fig. 3(a-c) shows the comparison between the two bosonic approaches (LSW theory and GA approach) with the exact solution for a moderate quench at a field $\Omega = 3\Omega_c$ . We observe that both the LSW and the GA approach correctly predict the light-cone structure of correlations, with the aperture of the light-cone reflecting the speed of the fastest quasi-particle excitations. Yet only the GA approach correctly captures the structure of correlations within the light-cone, exhibiting damped oscillations of the correlations at each distance after the correlation front has passed; while LSW theory predicts undamped oscillations.

A global picture on the correlations developing at long times can be gathered by looking at the time-averaged structure factor. We first introduce the instantaneous structure factor, namely the Fourier transform of the spin-spin correlation function:

$$S_k^{\mu\mu}(t) = \frac{1}{N} \sum_{i,j} e^{ik \cdot (r_i - r_j)} C^{\mu\mu}(i,j;t) \,. \tag{17}$$

Then we shall examine its time average for $\mu = z$, $\overline{S_k^{zz}} = \frac{1}{N\tau} \int_0^\tau dt \, S_k^{zz}(t)$. Fig. 3(d) shows this quantity as predicted by the GA approach, and compared with LSW theory and the exact

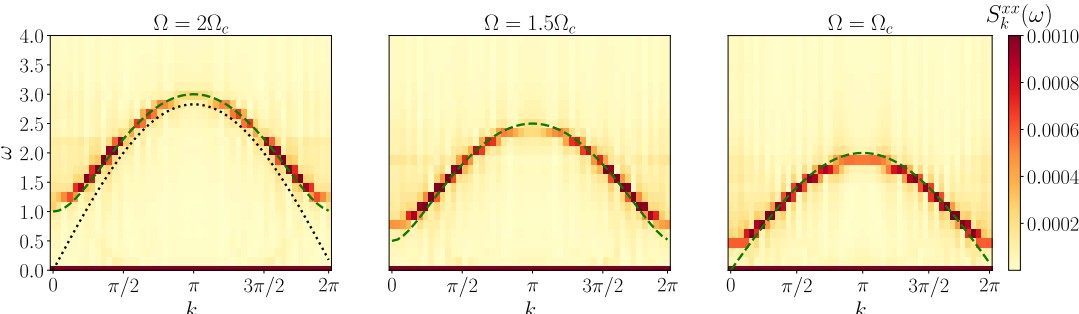

Figure 5: Quench-spectroscopy spectral function $S_k^{xx}(\omega)$ extracted from the quench dynamics of a 1d TFIM with $N = 50$ for three field values $\Omega = 2\Omega_c, 1.5\Omega_c$ and $\Omega_c$. The dotted black line in the leftmost panel corresponds to $2\omega_{\mathbf{k}}$ as predicted by LSW theory; while the green dashed line (in all panels) stands for $2\epsilon_k$ from the exact solution. The time window for the Fourier transform is $\tau J = 40$.

result. The agreement of the GA prediction with the exact result is rather remarkable, while LSW theory is off by almost a factor of 2, reflecting the presence of undamped revivals of correlations that we already commented on above.

### 3.2.2 $\langle S^x S^x \rangle$ correlations and quench spectroscopy

The spatio-temporal pattern of correlations establishing in the system during the dynamics is fundamentally dictated by the spectral features of the excitations in the system. Indeed it not only reveals the maximum group velocity via the aperture of the light cone, but it can also reveal the full dispersion relation when inspecting the spatial structure within the light cone. This insight is clearly suggested by LSW theory, when considering the evolution of the instantaneous structure factor – namely the spatial Fourier transform of the correlation pattern at time $t$, as in Eq. (17). In particular, for $\mu = x$, LSW predicts that $S_k^{xx}(t)$ oscillates at a frequency $2\omega_k$ (around a constant term), where $\omega_k$ is the dispersion relation of Eq. (13) [14, 46]; therefore its time-like Fourier transform reveals the full dispersion relation.

More generally we can introduce the Fourier transform of the instantaneous structure factor

$$S_{\mathbf{k}}^{xx}(\omega) = \int_{-\infty}^{+\infty} \frac{dt}{2\pi} e^{-i\omega t} S_k^{xx}(t) \tag{18}$$

$$= \frac{1}{4} \sum_{m,n} \langle \psi(0)|m\rangle \langle n|\psi(0)\rangle \, \delta(\omega - \omega_{n,m}) \big( \langle m| \big( \hat{S}_{\mathbf{k}}^+ \hat{S}_{-\mathbf{k}}^+ + \text{h.c.} \big) |n\rangle + \langle m| \big( \hat{S}_{\mathbf{k}}^+ \hat{S}_{\mathbf{k}}^- + \hat{S}_{-\mathbf{k}}^- \hat{S}_{-\mathbf{k}}^+ \big) |n\rangle \big),$$

where $\omega_{n,m} = \omega_n - \omega_m$; $|m\rangle, |n\rangle$ are Hamiltonian eigenstates; and we have introduced the Fourier transformed spin operator

$$\hat{S}_{\mathbf{k}}^- = \frac{1}{\sqrt{N}} \sum_i e^{i\mathbf{k}\cdot\mathbf{r}_i} \hat{S}_i^-, \qquad \hat{S}_{\mathbf{k}}^+ = (\hat{S}_{-\mathbf{k}}^-)^\dagger, \tag{19}$$

which create (resp. destroy) a spin flip delocalized throughout the system with momentum $\mathbf{k}$.

This spectral function contains therefore frequency contributions coming from transitions between pairs of states $|n\rangle, |m\rangle$ which are connected by the creation (resp. destruction) of two delocalized spin flips at wavevectors $\mathbf{k}$ and $-\mathbf{k}$ via the operators $\hat{S}_{\mathbf{k}}^- \hat{S}_{-\mathbf{k}}^-$ (resp. $\hat{S}_{\mathbf{k}}^+ \hat{S}_{-\mathbf{k}}^+$); or by the successive creation and destruction of such a spin flip (via the operator $\hat{S}_{\mathbf{k}}^+ \hat{S}_{\mathbf{k}}^-$). The pairs of states entering in the spectral function are weighted by their overlap with the initial state $|\psi(0)\rangle$; if this state has a significant overlap with the ground state $|n\rangle = |0\rangle$, the spectral

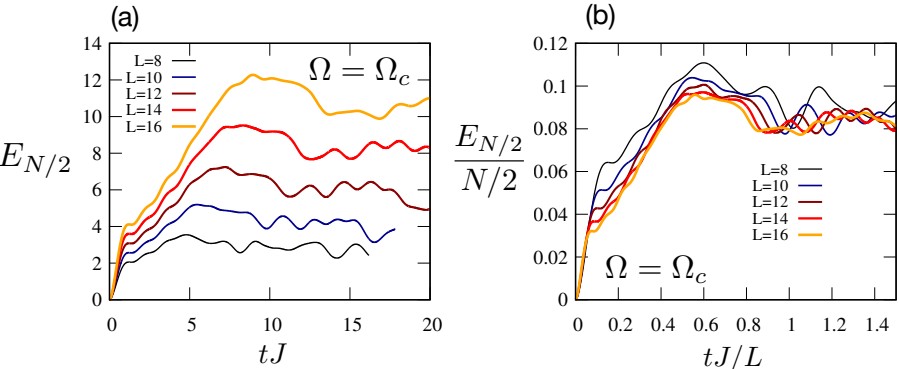

Figure 6: Evolution of the half-system entanglement entropy $E_{N/2}$ following a quench to $\Omega = \Omega_c$ in the 2d TFIM, obtained via the GA approach: (a) entropy evolution for various lattice sizes $N = L \times L$; (b) entropy per spin as a function of the time rescaled by the linear dimension $L$ of the system.

function will receive contributions from transitions $|0\rangle \rightarrow |m\rangle$ induced by the creation of two elementary excitations at opposite wavevectors $\pm\mathbf{k}$. When considering quenches for $\Omega \geq \Omega_c$, the ground state of the TFIM is paramagnetic and strongly aligned with the applied field, so that we can imagine that these elementary excitations correspond to the creation of two quasiparticles at opposite wavevectors. In the specific case of the 1d TFIM, the transition frequency $\omega_{m,n}$ should therefore correspond to the energy $\epsilon_k + \epsilon_{-k} = 2\epsilon_k$. These considerations suggest that the double (spatial+temporal) Fourier transform of the $C^{xx}$ correlation function should reveal the dispersion relation of the elementary excitations: this insight is at the basis of the so-called *quench spectroscopy* (QS), already proposed in Refs. [14–16].

Fig. 5 shows the QS spectral function $S_k^{xx}(\omega)$ as obtained via the GA approach for three values of the transverse field $\Omega$ ($2\Omega_c$, $1.5\Omega_c$ and $\Omega_c$). For all three cases we clearly observe a strong signal, suggesting the ability of the QS scheme to reconstruct the dispersion relation of elementary excitations. In the case of the $C^{xx}$ correlation function and of its Fourier transform, we do not easily have access to the exact result, given that the $\hat{S}_i^x \hat{S}_j^x$ operator, when expressed in terms of the free fermions, contains a string of operators associated with all the sites between the $i$th and the $j$th one. Yet we can compare the predictions of the GA approach with the dispersion relation $2\epsilon_k$ obtained via the Jordan-Wigner transformation, Eq. (15). The first panel of Fig. 5 shows this comparison for $\Omega = 2\Omega_c$, along with the prediction of the dispersion relation from LSW theory, Eq. (13). Clearly there is excellent agreement between the dispersion relation reconstructed via QS within the GA approach and the exact prediction; while the LSW dispersion differs from the fermionic one, predicting a gapless dispersion mode (as the field in question corresponds to the instability field $\Omega^*$ for LSW theory). The agreement between the GA prediction via QS and the exact dispersion relation persists also at lower fields (down to $\Omega_c$) for what concerns the dispersion relation at moderate to high frequencies; while the GA prediction misses the fact that the dispersion relation becomes exactly gapless for $\Omega_c$. This may be due to the Gaussian-state approximation, but also to the truncation of the bosonic nonlinearities to quartic order, altering the nature of the Hamiltonian of the system, and the position of its critical point.

### 3.3 Discussion

In this section we have seen that the GA approach is able to quantitatively capture many aspects of the quench dynamics in the 1d TFIM for large fields $\Omega$, and down to the critical one $\Omega_c$. This means that a Gaussian-state Ansatz for strongly interacting HP bosons is able to

mimic the physics of free spinless fermions onto which the one-dimensional spin model can be exactly mapped. On the other hand a purely linear theory (the LSW one) only works at very large fields, while it fails at intermediate ones due to a dynamical instability. Hence the success of the GA approach entirely relies on its ability to account (at least partially) for the nonlinearities of the bosonic physics.

In particular we observed that the dynamics of correlations allows one to reconstruct the dispersion relation for elementary excitations via Fourier transformation (quench spectroscopy, QS); and that the GA results for the quench-spectroscopy signal are in very good agreement with the dispersion relation expected for the elementary fermionic excitations in the 1d TFIM. A word of caution is in order nonetheless when considering the QS signal at low fields $\Omega \approx \Omega_c$, in spite of the (partial) agreement between the GA predictions and the exact dispersion relation. In that regime LSW fails completely, suggesting that the picture of elementary excitations as being bosonic spin flips may no longer be valid. Indeed the picture of elementary excitations offered by the Jordan-Wigner mapping is that of fermionic quasi-particles, namely of Bogolyubov transformations of the $\hat{f}_i, \hat{f}_i^\dagger$ operators, which in turn do not correspond to simple spin-flip operators, but to spin operators "dressed" with operator strings, $\hat{f}_i^\dagger = \hat{S}_i^- \exp[i\pi \sum_{j<i}(1/2 - \hat{S}_j^z)]$. Hence in general the QS spectral function, which probes transitions between eigenstates generated by spin operators such as $\hat{S}_k^- \hat{S}_{-k}^-$, may not be able to reconstruct sharp elementary fermionic excitations, but rather a continuum thereof – similar to other spectral probes coupling to spin operators, such as neutron scattering on magnetic insulators [34]. Truncating the nonlinearity of the bosonic Hamiltonian to quartic order and studying the system within a GA approach, one seemingly observes sharp features in the QS spectral function, which nonetheless may not persist as such in the exact solution to the problem. Nonetheless it is remarkable that the QS scheme may allow for the reconstruction of crucial features of the free-fermion dispersion relation down to the critical field.

# 4 TFIM on a square lattice

We now apply the GA approach to a more generic system, namely the (non-integrable) TFIM on a square lattice. For its ground-state and equilibrium physics, this model lends itself to numerically exact quantum Monte Carlo (QMC) calculations, which predict a ferromagnetic-to-paramagnetic quantum phase transition at zero temperature for $\Omega_c = 1.52219(1)J$ [47]. LSW theory built around the state polarized with the field is expected to be predictive for large fields, but it becomes unstable for $\Omega < \Omega^* = 2J$. As we shall see in the following, the GA approach is instead stable for all fields. In order to test the GA predictions for the non-equilibrium dynamics, we shall compare them with the results of time-dependent variational Monte Carlo (tVMC) based on the convolutional neural network (CNN) wavefunction [33]; the latter approach reproduces as well the results from a tensor-network Ansatz for the evolved wavefunction, and therefore its predictions can be considered as offering a very trustworthy reference. We would like to point out that the CNN wavefunction calculations, while very powerful, are extremely demanding computationally (when pushed to system sizes $N = 10 \times 10$), as documented in great details by Ref. [33]. On the other hand the GA calculations on the same system sizes with periodic boundary conditions only take a few tens of seconds on a standard laptop. Concerning the spectral properties, we shall use for comparison the results of a high-order series expansion around the $J \to 0$ limit, providing the most accurate estimate for the dispersion relation of the elementary excitations [48].

We shall start our discussion from the evolution of entanglement entropies; and then, similarly to the case of the 1d TFIM, we shall discuss the evolution of the transverse magnetization and of the correlation pattern revealing the quasi-particle spectrum of the system.

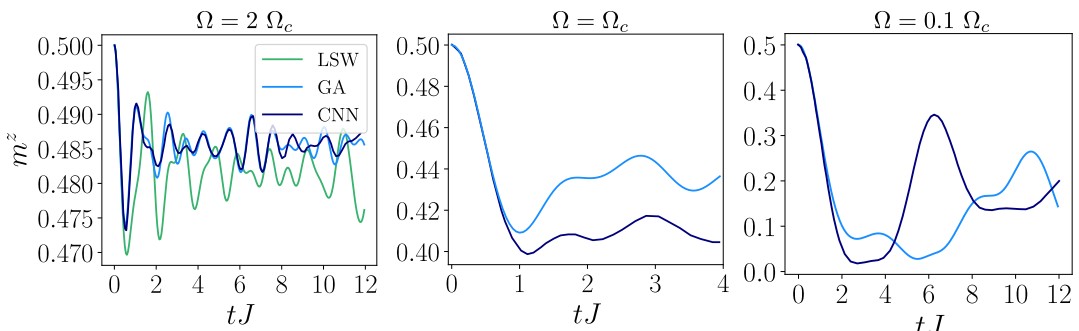

Figure 7: Evolution of the transverse magnetization $m^z$ of the 2d TFIM after a quantum quench to three field values $\Omega = 2\Omega_c$, $\Omega_c$ and $0.1\Omega_c$. All the data refer to a lattice of $N = 10 \times 10$ spins with periodic boundary conditions. The leftmost panel compares the GA results with the LSW and the ones based on the CNN wavefunction of Ref. [33]; the other panels only report the GA and the CNN results.

## 4.1 Entanglement entropy

### 4.1.1 Quantum relaxation and entanglement spreading

The spreading of entanglement is at the core of the process of relaxation of a quantum system when driven away from equilibrium. Indeed in quantum systems the emergence of statistical ensembles upon equilibration is expected to happen at the level of the pure-state vector $|\psi(t)\rangle$ describing the evolved system, when the state is looked at locally through its "marginals", namely the reduced state of a subsystem $A$ comprising only a fraction of the sites of the lattice

$$\hat{\rho}_A(t) = \mathrm{Tr}_B |\psi(t)\rangle\langle\psi(t)| . \tag{20}$$

Here $\mathrm{Tr}_B$ indicates the partial trace of the projector onto the evolved state with respect to the degrees of freedom hosted by the complement $B$ of the subsystem $A$ of interest. If subsystem $A$ is weakly coupled to its complement – namely the energy associated with the Hamiltonian terms $\hat{\mathcal{H}}_{AB}$ coupling $A$ and $B$ is much smaller than that of the Hamiltonian $\hat{\mathcal{H}}_A$ describing the energy of the isolated subsystem – then $\hat{\rho}_A(t)$ at sufficiently long times is expected to reproduce the density matrix of the equilibrium ensemble of the system, conditioned on the values of the quantities conserved by the dynamics [8, 9]. If the energy is the only conserved quantity, then one expects that $\hat{\rho}_A$ reproduces the Gibbs ensemble at long times, namely $\hat{\rho}_A(t \to \infty) \approx \exp(-\hat{\mathcal{H}}_A/T)/\mathcal{Z}_A$ for some temperature $T$ (dictated by the initial state of the system), where $\mathcal{Z}_A = \mathrm{Tr}[\exp(-\hat{\mathcal{H}}_A/T)]$.

The equilibrium ensemble is generically associated with a finite entropy, meaning that $\hat{\rho}_A(t)$ is a mixed state, in spite of the fact that the overall state of the system remains pure. The mixedness of the reduced state of a subsystem is a consequence of the joint state of the $A$ and $B$ subsystems becoming entangled (namely inseparable) [49], and it can be quantified using the von Neumann entanglement entropy

$$E_A = -\mathrm{Tr}[\hat{\rho}_A \ln \hat{\rho}_A] . \tag{21}$$

### 4.1.2 Entanglement entropies from the Gaussian Ansatz

The entanglement entropy $E_A$ is not a simple observable, and in general it is rather hard to extract from a many-body calculation. Nonetheless the GA approach provides a very direct access to entanglement entropies given that, by construction, the reduced state $\hat{\rho}_A$ is postulated

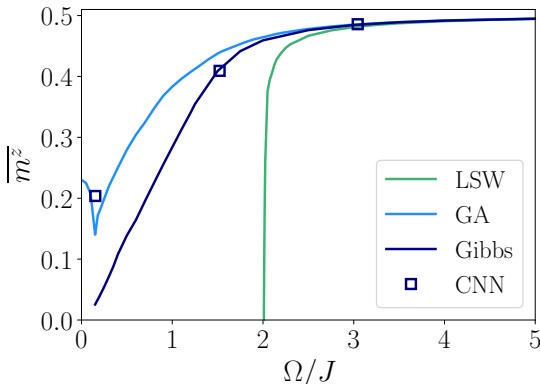

Figure 8: Time-averaged transverse magnetization along the quench dynamics of the 2d TFIM, comparing the GA, LSW and CNN predictions. The time window for the average is $\tau J = 20$ for the GA and LSW results while it is the one explored in Ref. [33] for the CNN results (the average is taken over the time interval $[1,4]J^{-1}$ for $\Omega = \Omega_c$ and $[1,12]J^{-1}$ for the other values of the field). All the other parameters are as in Fig. 7. The results corresponding to the Gibbs ensemble are obtained via QMC on a $N = 8 \times 8$ lattice for a temperature matching the energy of the initial state, as mapped in Ref. [20].

to be a Gaussian state, whose properties are all stored in the covariance matrix, including its entropy. The GA approach implies that at all times the reduced state is the exponential of a quadratic form in the boson operators

$$\hat{\rho}_A(t) =: \frac{1}{\mathcal{Z}_A} \exp[-\mathcal{H}_E(t)], \tag{22}$$

with the so-called entanglement Hamiltonian being given by

$$\mathcal{H}_E(t) = \frac{1}{2} \sum_{i,j \in A} \left[ \mathcal{Q}_{ij}(t) \, \hat{b}_i \hat{b}_j + \mathcal{P}_{ij}(t) \, \hat{b}_i^\dagger \hat{b}_j + \text{h.c.} \right], \tag{23}$$

where $\mathcal{Q} = \{\mathcal{Q}_{ij}\}$ and $\mathcal{P} = \{\mathcal{P}_{ij}\}$ are $N_A \times N_A$ time-dependent matrices. Here we consider that $\langle \hat{b}_i \rangle = 0$ at all times, as it is the case for the evolutions studied in this work – otherwise the entanglement Hamiltonian contains a linear term as well. Eq. (22) has the form of the Gibbs state of a quadratic bosonic Hamiltonian, whose thermodynamics is exactly solvable via a Bogolyubov transformation $\hat{b}_i = \sum_a \left( u_i^{(a)} \hat{c}_a + v_i^{(a)} \hat{c}_a^\dagger \right)$ to new bosonic operators $\hat{c}_a, \hat{c}_a^\dagger$, which diagonalizes the entanglement Hamiltonian:

$$\hat{\rho}_A(t) = \frac{1}{\mathcal{Z}_A} \exp\left( -\sum_a e_a \hat{c}_a^\dagger \hat{c}_a \right), \tag{24}$$

where $a$ is the index of the entanglement Hamiltonian eigenmodes. The entanglement entropy is therefore the entropy of this free-boson system, taking the form

$$E_A = \sum_a [(1 + n_a) \ln(1 + n_a) - n_a \ln n_a], \tag{25}$$

where $n_a = (\exp(e_a) - 1)^{-1}$ is the Bose occupation factor of the $a$-th eigenmode.

The $u$ and $v$ coefficients of the Bogolyubov transformation diagonalizing the density matrix can be obtained by the diagonalization of the reduced covariance matrix for the subsystem $A$ [50]. For a subsystem comprising $N_A$ sites, we introduce the $2N_A \times 2N_A$ matrix

$$\mathcal{U}_A = \begin{pmatrix} U_A & V_A^* \\ V_A & U_A^* \end{pmatrix}, \qquad (26)$$

where $U_A = \left(\mathbf{u}^{(a=1)}...\mathbf{u}^{(a=N_A)}\right)$ and $V_A = \left(\mathbf{v}^{(a=1)}...\mathbf{v}^{(a=N_A)}\right)$ are $N_A \times N_A$ matrices built from the $\mathbf{u}^{(a)} = (u_1^{(a)},...,u_{N_A}^{(a)})^T$ and $\mathbf{v}^{(a)} = (v_1^{(a)},...,v_{N_A}^{(a)})^T$ column vectors, with the property of Bogolyubov-diagonalizing the quadratic entanglement Hamiltonian of Eq. (23):

$$\mathcal{U}_A \begin{pmatrix} \mathbb{1}_{N_A} & {}_{N_A} \\ {}_{N_A} & -\mathbb{1}_{N_A} \end{pmatrix} \begin{pmatrix} \mathcal{Q} & \mathcal{P} \\ \mathcal{P}^\dagger & \mathcal{Q} \end{pmatrix} \mathcal{U}_A^{-1} = \mathrm{diag}(e_1,...,e_{N_A},-e_1,...,-e_{N_A}). \qquad (27)$$

Then the $\mathcal{U}_A$ matrix can be obtained from the knowledge of the Green's functions, namely the $N_A \times N_A$ matrices $G^{(A)} = \{G_{ij}\}_{i,j\in A}$ and $F^{(A)} = \{F_{ij}\}_{i,j\in A}$, as it diagonalizes the matrix [50]:

$$\begin{pmatrix} -\mathbb{1}_{N_A} - (G^{(A)})^* & F^{(A)} \\ -(F^{(A)})^* & G^{(A)} \end{pmatrix} = \mathcal{U}_A \begin{pmatrix} \mathrm{diag}(-1-n_a) & {}_{N_A} \\ {}_{N_A} & \mathrm{diag}(n_a) \end{pmatrix} \mathcal{U}_A^{-1}. \qquad (28)$$

### 4.1.3 Results for the 2d TFIM

Fig. 6 shows the GA prediction for the time evolution of the entanglement entropy for a $N = L \times L$ lattice with periodic boundary conditions, where the subsystem $A$ corresponds to a half torus with $N_A = L \times L/2 = N/2$ sites. Fig. 6(a) shows the entanglement dynamics following a quench to the critical field $\Omega = \Omega_c$ for various system sizes: for each size one clearly observes an initial ramp-up phase, followed by a phase displaying fluctuations around a seemingly equilibrated value. When rescaling the entanglement entropy by the subsystem size $N/2$, and the time axis by the linear size $L$ of the system, one can bring the curves to approximately collapse. This demonstrates that the equilibration time scales linearly with the linear dimensions of the subsystem, as expected if the entanglement spreading is driven by the ballistic propagation of quasi-particle excitations; and that the regime of equilibration corresponds to a density matrix exhibiting extensive entropy, as expected *e.g.* for the Gibbs ensemble. The fact that the GA can reach extensive entanglement entropies is rather unsurprising, given that the entanglement entropy corresponds to the thermal entropy of a quadratic bosonic theory. This clearly suggests that, unlike the case of other Ansätze [17, 18], the entanglement content of the GA is rather arbitrary; and that its limitations come mostly from its simplifying assumptions on the statistics of quantum fluctuations. Whether the state towards which the GA dynamics evolves at long times corresponds to thermal equilibrium described by the Gibbs ensemble will be subject of the following section.

## 4.2 Transverse magnetization

We now examine the dynamics of the transverse magnetization $m^z = N^{-1} \sum_i \langle S_i^z \rangle$, which we shall use to probe the question of thermalization along the non-equilibrium dynamics. Fig. 7 shows the evolution of the magnetization for three values of the field ($\Omega = 2\Omega_c$, $\Omega_c$ and $0.1\Omega_c$) comparing the GA results on a $10 \times 10$ lattice with the prediction of the CNN wavefunction (from Ref. [33]); in the case of the larger field $\Omega = 2\Omega_c > \Omega^*$ we can also compare with LSW theory. For this latter field we observe that the agreement between the GA results and the CNN ones is rather remarkable; and that it is fundamentally due to the inclusion of the quartic nonlinearities in the bosonic theory, since the LSW results, which ignore all nonlinearities, deviate significantly from both the GA and the CNN ones. The agreement between GA and

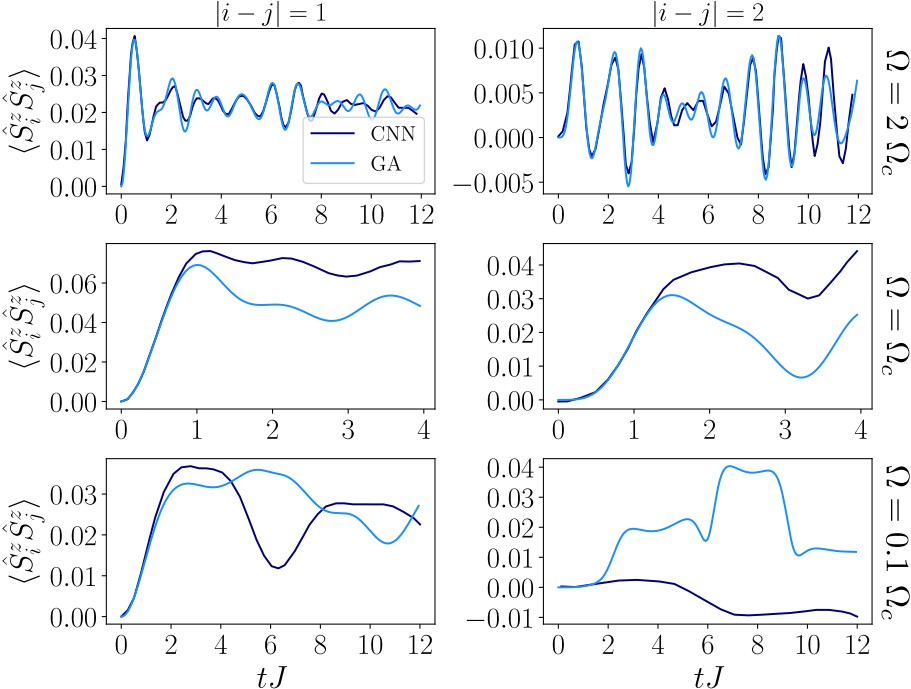

Figure 9: Evolution of the spin-spin correlation function for the $z$ component in the 2d TFIM, at two distances $|i-j| = 1$ (first column) and $|i-j| = 2$ second column, for three field values (for each of the three rows) $\Omega = 2\Omega_c, \Omega_c$ and $0.1\Omega_c$. All the panels compare the GA results with the CNN ones from Ref. [33], and have been obtained for a $N = 10 \times 10$ lattice.

CNN Ansatz is less good for $\Omega = \Omega_c$, albeit the most salient features of the evolution (such as the characteristic timescales for the evolution of the magnetization) are correctly captured by the GA. On the other hand for $\Omega = 0.1\Omega_c$ the agreement between GA and CNN is only in the overall amplitude of the magnetization fluctuations.

Fig. 8 addresses the question of thermalization of the many-body dynamics, by showing the comparison between the time-averaged magnetization $\overline{m^z}$ as obtained within the GA approach; the same quantity obtained from the CNN Ansatz (with results averaged over the limited time window studied in Ref. [33]); and the predictions of the Gibbs ensemble for a $N = 8 \times 8$ system. The Gibbs-ensemble results have been obtained via quantum Monte Carlo (based on the stochastic series expansion [51] approach) at a temperature $T$ such that $\langle \hat{\mathcal{H}} \rangle_T = \langle \psi(0) | \hat{\mathcal{H}} | \psi(0) \rangle$. We have used the temperatures corresponding to the expected Gibbs ensemble for varying $\Omega$ as mapped out in Ref. [20]. We clearly observe that for $\Omega \gtrsim 2J$ the time-averaged magnetization reproduces rather closely the Gibbs ensemble result. On the other hand the time-averaged GA results and the Gibbs-ensemble prediction deviate from each other significantly at lower fields, with the dynamical result systematically overestimating the thermal equilibrium one. This is clearly *not* a limitation of the GA approach, as a similar result holds as well for the limited CNN data offered in Ref. [33]. This apparent lack of thermalization at low fields (at least over the limited time window of interest) can be naturally understood in the limit $\Omega = 0$, which, as already discussed above, corresponds to an integrable case in which all $S_i^x$ operators commute with the Hamiltonian. Therefore it is not entirely surprising that a non-thermalizing dynamics sets in when $\Omega \to 0$ – this is apparently similar to the case of the 1d TFIM discussed above, although the 1d TFIM is provably integrable for *any* value of $\Omega$. Hence we conclude that the non-thermal behavior of the averaged magnetization observed at low fields within the GA – in particular, the fact that the magnetization does not relax towards a

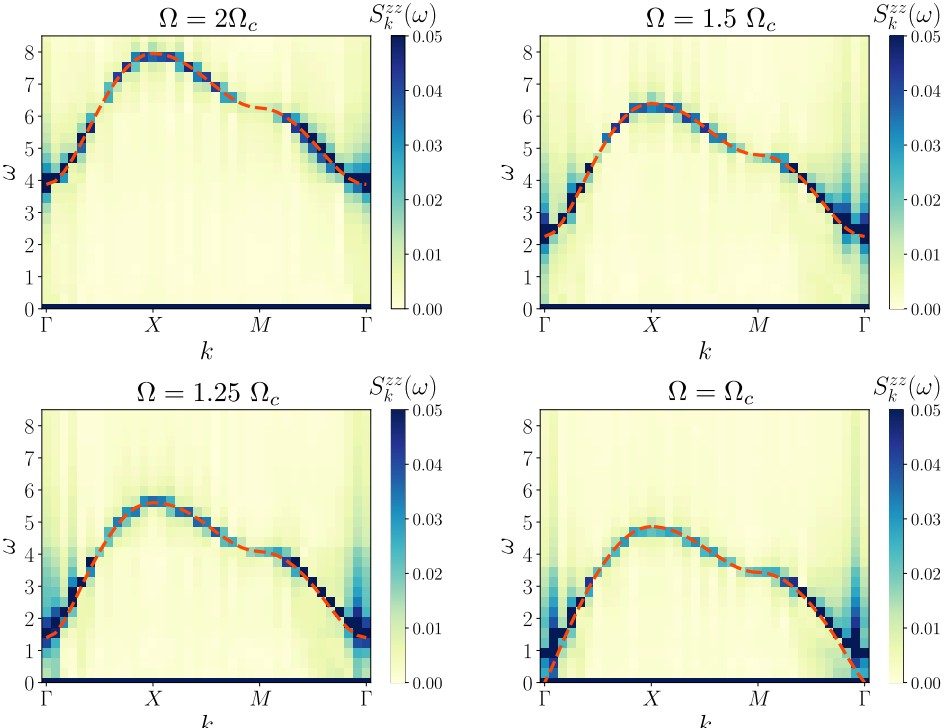

Figure 10: Quench-spectroscopy spectral function $S_k^{xx}(\omega)$ extracted from the quench dynamics of a 2d TFIM with $N = 24 \times 24$ for four field values $\Omega = 2\Omega_c, 1.5\Omega_c, 1.25\Omega$ and $\Omega_c$. The $x$ axis refers to a linear trajectory in the 2d Brillouin zone, going from $\Gamma = (0,0)$ to $X = (\pi, \pi)$ to $M = (\pi, 0)$ and then back to $\Gamma$. The dashed red line in all panels stands for $2\varepsilon_{\mathbf{k}}$ from the series expansion of Ref. [48]. The Fourier transform is performed over the time window $\tau J = 25$.

vanishing time-averaged value when $\Omega \to 0$ – may be in fact an actual feature of the dynamics, and not at all a limitation of the GA approach.

## 4.3 Correlation dynamics and quench spectroscopy

We complete our study by considering the dynamics of correlations, and how it can reveal fundamental features of the excitation spectrum. Fig. 9 shows the GA and CNN predictions for the evolution of the $C^{xx}(i, j; t)$ correlation function for two inter-site distances $|i - j| = 1$ and 2, and for the three values of the field already explored above ($\Omega = 2\Omega_c$, $\Omega_c$ and $0.1\Omega_c$). Similar to the case of the magnetization, the agreement between GA and CNN results is rather remarkable for the largest field, while it remains acceptable for $\Omega = \Omega_c$ and it degrades rather drastically for $\Omega = 0.1\Omega_c$. This suggest therefore that the GA predictions remain quantitative at least over the entire field range $\Omega \gtrsim \Omega_c$.

A more global picture of the correlation dynamics is offered by the quench-spectroscopy analysis of the $C^{xx}(i, j; t)$ correlation function. As already discussed in Sec. 3.2.2, this analysis is expected to reveal the dispersion relation of elementary excitations at least in the paramagnetic phase for $\Omega > \Omega_c$, in which elementary excitations should be generated by spin flips along the $z$ axis, induced by the $\hat{S}_i^x$ operators. Unlike the case of the 1d TFIM, for the 2d system the bosonic picture of the elementary excitations can be expected to be valid at all fields, and therefore the GA predictions for the results of the QS analysis may be remain quantitative down to low fields.

In order to benchmark the GA results we make use of the most accurate predictions for the dispersion relation to our knowledge, namely the ones resulting from a series expansion in powers of $J/\Omega$ around the limit $J \to 0$ up to order 4 [48]. The QS spectral function for the 2d TFIM is shown in Fig. 10 for four field values ($\Omega = 2\Omega_c, 1.5\Omega_c, 1.25\Omega_c$ and $\Omega_c$) on approach to the critical point, and compared with $2\varepsilon_{\mathbf{k}}$ where $\varepsilon_{\mathbf{k}}$ is the dispersion relation obtained from the series expansion. The agreement between the GA results and the series-expansion predictions is rather striking, bearing two consequences: 1) the fact that the GA approach can faithfully reconstruct the nonlinear excitation spectrum of a strongly interacting system (as a reminder, the linear excitation spectrum develops imaginary frequencies for $\Omega < \Omega^* \approx 1.3\Omega_c$); and 2) the fact that the correlation dynamics far from equilibrium is sensitive to the existence of a ground-state phase transition, as the QS function exhibits the softening of the excitation spectrum upon approaching the quantum critical field. The fact that the QS signal from the GA approach does not become fully gapless at the exact critical field might be due to the GA approximation to the evolved state; as well as to the fact that the bosonic Hamiltonian differs from the exact spin one because of the truncation of nonlinearities to quartic order.

## 5 Conclusions and outlook

We have introduced and validated the Gaussian-Ansatz (GA) approach to the non-equilibrium dynamics of quantum spin systems. Our approach is based on the Holstein-Primakoff (HP) mapping of the spin model onto a nonlinear bosonic one, and to the truncation of the nonlinear bosonic Hamiltonian to a finite order. The Gaussian-state Ansatz allows for the efficient calculation of the non-equilibrium dynamics: all the information on the Gaussian state of an $N$-spin system is contained in the $\mathcal{O}(N^2)$ elements of the covariance matrix, circumventing the exponential growth of the Hilbert space dimensions. The quantization axis defining the HP transformation must be chosen so that the initial state of the evolution is a Gaussian state of the HP bosons – in this work we have investigated the case of dynamics initialized in a coherent spin state aligned with the local quantization axis, so that the initial bosonic state corresponds to the vacuum. The ensuing dynamics is expected to preserve the Gaussian nature of the state as long as the density of bosons, proliferating under the non-equilibrium evolution, remains moderate. More general initial states can be explored as well – *e.g.* the ground states, or even the low-temperature equilibrium states, of quantum spin Hamiltonians approximated via the linear or non-linear (*i.e.* modified) spin-wave theory.

Accounting for nonlinearities turns out to be essential in order to keep the proliferation of HP bosons under control during the dynamics; and to reproduce the emergence of the equilibrium statistical averages of local observables in the long-time dynamics of the system. We investigated the quench dynamics of the transverse-field Ising model in one and two dimensions starting from a state aligned with the field, and we observed that the predictions of the GA approach remain quantitative for quenches to fields down to the critical field associated with the ground-state paramagnetic-to-ferromagnetic transition. In particular the GA predictions for the evolution of the spin-spin correlations allow us to reconstruct the dispersion relation of elementary excitations via a quench-spectroscopy analysis (Fourier transform of the spatio-temporal correlation pattern): the dispersion relation resulting from the GA is in very good agreement with the most accurate predictions for the elementary excitation spectrum of the model of interest. In particular the resulting excitation spectrum exhibits mode softening upon approaching the quantum critical field, suggesting that the non-equilibrium dynamics of the system can reveal the existence of ground-state quantum critical points.

Our results show that the GA approach to the nonlinear bosonic equivalent of quantum spin Hamiltonians allows one to efficiently investigate the non-equilibrium dynamics of the system far beyond the linearized treatment, and to account for most salient effects of a) relaxation of local observables to an equilibrium (Gibbs or generalized Gibbs) ensemble; and b) the emergence of the dispersion relation of elementary excitations in the correlation dynamics. The main limitations of the GA approach to quantum spin systems stem from its approximate treatment of quantum fluctuations; and, equally importantly, from the fact that the nonlinear bosonic Hamiltonian needs to be truncated to a finite order in its otherwise infinite expansion in powers of $n/(2S)$ (where $n$ is the local boson density and $S$ the spin length). While in this work we chose to consider the minimal nonlinear Hamiltonian (truncated to quartic order), the expansion can be pushed to higher order, with the simple effect of complicating the derivation of the equations of motion for the elements of the covariance matrix. In particular, Ref. [52] introduces generalized spin-to-boson transformations which, depending on the spin length $S$, allow one to map the spin operators onto finite-order polynomials of bosonic operators – as an example, $S = 1/2$ operators can be cast as cubic polynomials. Using such transformations, one can aim at retaining all nonlinearities contained in the spin Hamiltonian.

The GA approach appears to be a very promising tool to investigate quantum spin dynamics far beyond the regimes explored in this work. First of all, the spin length $S$ enters in the equations of motion for the covariance matrix as a parameter, so that spins of arbitrary length – namely the quantum dynamics of systems of interacting *qudits* – can be studied without any additional computational cost resulting from the growth of the local Hilbert-space dimensions. This is particularly relevant when considering *e.g.* the dynamics of large-spin magnetic atoms trapped in optical lattices [25]. The ability of the GA approach to investigate the equilibration of the system suggests that the same approach can be used to investigate the failure of the system to relax in the presence of strong disorder, leading to many-body localization. The latter phenomenon, forcing the evolved state of the system to remain parametrically close to the initial state, appears to be very promising for a GA study, given that the GA is fully justified when the evolved state of the system remains close to the initial state if such a state is Gaussian. Further extensions of the GA approach may involve the study of dynamical phase transitions with Loschmidt echo singularities [53] (and the Loschmidt echo can be efficiently evaluated for Gaussian states [54]); the study of time crystallization under periodic driving [55]; and the extension of the approach to open-system dynamics, most importantly within a wavefunction Monte Carlo approach to the master-equation dynamics, in which each quantum trajectory can be approximated via a Gaussian state, as already done for models of strongly interacting photons [56]. Its very moderate computational cost (amounting at most to the numerical solution of $\mathcal{O}(N^2)$ coupled differential equations in the absence of any symmetry) makes of the GA approach a very promising tool to efficiently benchmark experimental quantum simulators in regimes (of very large system sizes, very large spins, etc.) in which other approaches – such as wavefunction-based Ansätze – become unpractical.

A way to systematically improve on the GA approach is to view it as one instance of approximation schemes involving the truncation of the cumulant expansion of $n$-point correlation functions. While the GA corresponds to neglecting cumulants of order three and higher, one can push the cumulant expansion to an order $n > 3$ at a cost scaling as $N^n$, as recently attempted *e.g.* in Ref. [57]. The possibility to systematically improve the GA approach by including higher-order cumulants at a polynomial cost invites one to view this scheme as a potential promising alternative to wavefunction based Ansätze for strongly interacting spin systems.

## Acknowledgments

We would like to thank H. Kurkjian, W. Verstraelen and M. Wouters for useful discussions. TR is supported by ANR ("EELS" project), QuantERA ("MAQS" project) and PEPR-Q ("QubitAF" project).

## A  Equations of motion for the transverse-field Ising model

Here we report the equations of motion for the covariance matrix when considering the bosonic quartic Hamiltonian of Eq. (10) onto which the transverse-field Ising model is mapped via the Holstein-Primakoff transformation.

$$\frac{\mathrm{d}}{\mathrm{d}t}G_{ij} = i\left(\mathcal{G}_{ij} - \mathcal{G}_{ji}^*\right), \tag{A.1a}$$

$$\frac{\mathrm{d}}{\mathrm{d}t}F_{ij} = i\left(\mathcal{F}_{ij} + \mathcal{F}_{ji} + j_{ij}^F\right), \tag{A.1b}$$

where $\mathcal{G}$, $\mathcal{F}$ and $j^F$ are $N \times N$ matrices. Their explicit forms are given by

$$\begin{aligned}
\mathcal{G}_{ij} = &-\frac{S}{2}\sum_k J_{ik}(G_{kj} + F_{kj}) \\
&-\frac{1}{8}\sum_k \left[J_{ik}(2G_{kk} + F_{kk}^*)F_{kj} + (2G_{ii} + F_{ii}^*)J_{ik}F_{kj}\right] \\
&-\frac{1}{8}\sum_k \left[J_{ik}(2G_{kk} + F_{kk})G_{kj} + (2G_{ii} + F_{ii}^*)J_{ik}G_{kj}\right] \\
&-\frac{1}{2}\sum_k J_{ik}\mathrm{Re}\left[G_{ki} + F_{ki}\right]G_{ij} - \frac{1}{4}\sum_k J_{ik}\left[G_{ki}^* + F_{ki}^*\right]F_{ij},
\end{aligned} \tag{A.2}$$

and

$$\begin{aligned}
\mathcal{F}_{ij} = &\frac{S}{2}\sum_k J_{ik}(G_{kj} + F_{kj}) - \Omega F_{ij} \\
&+\frac{1}{8}\sum_k \left[J_{ik}(2G_{kk} + F_{kk})G_{kj} + (2G_{ii} + F_{ii})J_{ik}G_{kj}\right] \\
&+\frac{1}{8}\sum_k \left[J_{ik}(2G_{kk} + F_{kk}^*)F_{kj} + (2G_{ii} + F_{ii})J_{ik}F_{kj}\right] \\
&+\frac{1}{2}\sum_k J_{ik}\mathrm{Re}\left[G_{ki} + F_{ki}\right]F_{ij} + \frac{1}{4}\sum_k J_{ik}\left[G_{ki} + F_{ki}\right]G_{ij},
\end{aligned} \tag{A.3}$$

where

$$j_{ij}^F = \frac{S}{2}J_{ij} + \frac{1}{4}\delta_{ij}\sum_k J_{jk}(G_{kj} + F_{kj}) + \frac{1}{8}J_{ij}(2G_{ii} + F_{ii} + 2G_{jj} + F_{jj}). \tag{A.4}$$

The linear spin-wave equations are recovered from these equations when discarding all non-linear terms.

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
