# Peer review of "Gaussian-state Ansatz for the non-equilibrium dynamics of quantum spin lattices"

_SciPost Physics, doi:SciPost Phys. 14, 151 (2023)_

## Round 2 · Referee Report · Anonymous · 2023-2-7

Strengths

1. Suggests a novel way to look at non-equilibrium dynamics of spin systems.
2. Presents very rigorous and accurate calculations in a very detailed and clear way (the analytics are explained and detailed far beyond the current average in similar papers, in my opinion).
3. Shows obvious benchmarks of known results in some regimes.
4. Discusses honestly the limitations and drawbacks of the proposed methods, in comparison with the successes; not hiding the weaknesses of the method is, in my opinion, a strength of the work.
5. Gives proper background and references.
6. Clearly written

Weaknesses

1. The method does not work any arbitrary regime (but the authors analyze and explain why in a convincing way).

Report

In their manuscript, Menu and Roscilde suggest to use a bosonic Gaussian state Ansatz (GA) for studying non-equilibrium dynamics of spin lattices. This is done through mapping the spin models to bosons using the Holstein-Primakoff approximation.

The authors employ the Gaussian ansatz for both a solvable model (transverse field Ising chain) and a non-solvable one (TFIM on a square lattice, where results using other methods - using Neural Networks - are available so comparison could be and was done as well).

In both cases the authors show in which regimes and for which quantities the method works well, and analyze what happens when it does not (and why). Studying dynamical properties, correlation functions and entanglement entropies, the authors demonstrate that in many cases their method is competitive compared with other numerical methods, which is remarkable and in a sense (positively) surprising given the use of Gaussian states which do not allow to describe well many important quantum properties and correlations. They clearly show for which quantities the method is valid and for which it is not, and explain well the relation between this and the expected influence of the method on quantum correlations (e.g. when entanglement entropy is computed).
By that I believe that SciPost's acceptance criterion 1 is met.

The mathematical rigour of this work is remarkable. Everything is written in full detail, and the readers can easily follow the derivations without having to fill in too many gaps (and definitely not very big ones) on their own. Besides that, the presentation and text are very clear, and so are the figures.

I do have two questions which may be of interest for the authors to comment on and/or think about (though I do not find it crucial for publication, since they refer to issues beyond the scope of the work):
1. If the mapping to bosons were done using Schwinger representations, which are exact, rather than the Holstein-Primakoff approximation, could this possibly significantly improve the accuracy of the results (e.g. into some other regimes too)? Is it possible to make some comment on that?
2. In the last few years, there have been some works by Tao Shi et al on the "non Gaussian formalism" where Gaussian states are extended to describe interacting physics. Are the authors familiar with these works? These may be useful in such contexts too.

Requested changes

Very minor things:
1. In Eq. (1), the "i<j" subscript under the sum is restricted to 1-D.
2. Is the appendix necessary? It is extremely short and could be easily incorporated into the main text.

---

## Round 2 · Referee Report · Anonymous · 2023-3-13

Strengths

-New ansatz for the dynamics of many-body spin systems.
-Complete study: background, strength, and weaknesses.
-The analysis gives physical meaning to the data and provides the reader with a clear intuition about the mechanism of the method.
-The manuscript is clear and well written.

Weaknesses

-none

Report

The authors rigorously study the approximated out-of-equilibrium dynamics of many body systems when only the value of the average field and the covariance matrix of the state are tracked. This is called the Gaussian ansatz. They adapt this method to spin models using the Holstein-Primakoff spins to boson mapping.

The authors benchmark the ansatz against solvable models and non-solvable models carefully analysing in which regimes the method works and for which quantities the method is able to appropriately describe the dynamics.

Remarkably the authors study in detail the weaknesses of the method and in which cases the methods fail, giving an explanation of the causes and thus allowing the reader to understand deeply the nature of the approximation imposed with the ansatz.

The manuscript is rigorous, clear, and very well written. The authors guide the reader patiently through all the part of the work allowing the reader to clearly understand the importance of their results, the strength of their results, and the weakness of their results. The attentive description of the background and the context in which this work lies, together with the thoughtful choice of references, allows for this work to be not only an excellent presentation of the author's novel results, but also a valuable reference work for the field.

Requested changes

-Reference [20] please add DOI: 10.1038/srep38185

-Reference [31] please add DOI: 10.3254/978-1-61499-526-5-169

---

## Editorial Decision

published